# The Korea Cancer Big Data Platform (K-CBP) for Cancer Research

**DOI:** 10.3390/ijerph16132290

**Published:** 2019-06-28

**Authors:** Hyo Soung Cha, Jip Min Jung, Seob Yoon Shin, Young Mi Jang, Phillip Park, Jae Wook Lee, Seung Hyun Chung, Kui Son Choi

**Affiliations:** 1Cancer Big Data Center, National Cancer Control Institute, National Cancer Center, Goyang 10408, Korea; 2National Cancer Center Hospital, National Cancer Center, Goyang 10408, Korea; 3Department of Cancer Control and Policy, Graduate School of Cancer Science and Policy, National Cancer Center, Goyang 10408, Korea

**Keywords:** cancer data, big data platform, clinical cancer registry, de-identification

## Abstract

Data warehousing is the most important technology to address recent advances in precision medicine. However, a generic clinical data warehouse does not address unstructured and insufficient data. In precision medicine, it is essential to develop a platform that can collect and utilize data. Data were collected from electronic medical records, genomic sequences, tumor biopsy specimens, and national cancer control initiative databases in the National Cancer Center (NCC), Korea. Data were de-identified and stored in a safe and independent space. Unstructured clinical data were standardized and incorporated into cancer registries and linked to cancer genome sequences and tumor biopsy specimens. Finally, national cancer control initiative data from the public domain were independently organized and linked to cancer registries. We constructed a system for integrating and providing various cancer data called the Korea Cancer Big Data Platform (K-CBP). Although the K-CBP could be used for cancer research, the legal and regulatory aspects of data distribution and usage need to be addressed first. Nonetheless, the system will continue collecting data from cancer-related resources that will hopefully facilitate precision-based research.

## 1. Introduction

In clinical oncology, there is a major paradigm shift with the advent and dissemination of high-throughput genome sequencing technologies and molecular-targeted therapies. The subsequent explosion of biomedical data requires that cancer hospitals and laboratories be equipped with proper information technology capabilities, such as massive data storage and a high-capacity analysis platform, to address the accumulation of genomic and clinical data. Indeed, advances in information technology enable integration and analysis of large-scale biomedical data [1], and provide vital support for “precision medicine” that promises to deliver lifestyle modification and therapy tailored to the individual patient [2]. Now that this precision-based approach is a part of standard care in clinical oncology in some high-income countries, the need for a database or data warehouse (DW) to collect and store a wide range of medical, genomic, and lifestyle data cannot be overemphasized [3].

Initial versions of clinical DW simply stored raw data that were generated over the course of diagnosis and treatment without data editing [4,5,6,7], essentially ruling out the possibility of downstream analysis [8]. Since the early 2000s, many institutions collected large quantities of clinical data and constructed patient registries by extracting critical information such as outcome data [9]. Subsequent efforts were made to improve the quality of clinical data and track these data on a long-term basis [10,11]. Databases were built by collecting data from multiple institutions [12,13], and a system was created that could quickly extract source data to reflect the latest state of clinical data collection [14]. In addition, a web-based platform was developed based on the registry to actively utilize clinical data [15], followed by the addition of more sophisticated functions such as data comparison and analysis [16]. In this process, various security technologies for managing personal and sensitive information were applied. Most clinical databases developed after 2015 focus on the possibility of linkage across multiple databases. For example, it is necessary to build an integrated database linking a clinical database and a genome database [17], or to build a multi-institutional registry that combines clinical data from various institutions [18,19,20,21,22,23]. Moreover, web-based systems that collect external data were constructed [24,25]. The need for a platform with a large scale and range, due to the increasing amount and quality of collected data, mandated that researchers develop a big data platform, instead of a simple web-based platform [26,27].

These big data platforms can connect, analyze, and utilize diverse genomic, lifestyle, and environmental data needed for precision medicine, and they support a variety of clinical studies [28,29]. Although big data platforms were constructed from the databases of various institutions, and utilized in Korea and abroad, they have limitations for precision medicine in terms of data quantity, quality, and linkage [29,30].

In this paper, we introduce the Korea Cancer Big Data Platform (K-CBP), a multi-database framework that collects clinical, genomic, imaging, and biobank data using secure de-identification technology that allows for clinical research and practice in future research.

## 2. Materials and Methods

### 2.1. Data Sources

Data collected by the K-CBP consisted of (1) clinical data from 2001 to August 2018, (2) genomic data from April 2017 to August 2018, (3) biological sample data from 2002 to August 2018, and (4) cancer registration statistics data from the national cancer control initiatives from 1999 to August 2018. Clinical data were obtained from patients who visited the National Cancer Center (NCC) Hospital. Information on diagnosis and treatment was extracted from the electronic medical records (EMR) system. Information on genomic data was collected from targeted next-generation sequencing (NGS) analysis. Datasets of biological samples that included blood and tissue based on meta-information such as sample storage and storage locations were also collected from cancer screening examinees. Cancer registration statistics data generated from four distinct national cancer control activities conducted by the NCCI (namely, the Korea Central Cancer Registry, National Cancer Screening Program, Financial Aid Program for Cancer Patients, and Cancer Hospice and Palliative Care Program) were linked using an alternative key system independent of clinical cancer registries. Furthermore, “family history” and “social history” data based on survey results were additionally merged into the data. Data imported into the platform were de-identified and stored in an independent DW (IDW). All processes of DW generation were logged, which allowed for a newly updated version of DW to be explicitly characterized and deal with previous versions of a DW. Figure 1 shows an overview of the K-CBP holdings data.

### 2.2. De-Identification of Records

To protect patient privacy and ensure data safety, direct identifiers (ID), such as name and resident registration number present in healthcare data, need to be removed or replaced with alternative keys. Quasi-identifiers that can identify individuals when combining multiple items such as address, height, and weight were subjected to de-identification [31]. In the K-CBP, de-identification was performed by replacing the direct identifiers of source data with alternative patient keys before importing the data to K-CBP. Quasi-identifiers, which have the potential capability of identifying individuals when multiple features are combined, were deleted during the data provision process. The level of data deletion was determined depending on the level of the user’s authority. 

The alternative patient keys were generated by combining information such as institution source and registration order on the basis of personal information, as well as social security number, which is the unique identification number assigned to every legal citizen in Korea. The direct IDs used in the generation of the alternative patient keys were encrypted and stored in a closed system to minimize the possibility of external leakage. Only an honest broker who preprocessed the provided data can access the closed system. Alternative patient keys were configured to be changed randomly every time the dataset was generated so they could not be traced backward. 

### 2.3. Creation of Clinical Datasets and Cancer Registries

The process of screening clinical data and creating a clinical cancer registry is shown in Figure 2. To create and refine datasets that contain clinically relevant information, raw data need to be carefully examined, and only meaningful features need to be extracted. To this end, we chose clinical datasets from raw clinical data, and potentially meaningful features were derived from the medical literature. Then, a panel of clinical experts and potential data users selected datasets and relevant features for each clinical cancer registry based on the evidence in the literature, clinical significance, and their potential for linkage to external data sources such as personal lifelog information, which includes health determinants and health status. The selected features were then extracted from the EMR system and converted to terms that conformed to terminology standards such as the Systematized Nomenclature of Medicine Clinical Terms (SNOMED-CT), Logical Observation Identifiers Names and Codes (LOINC), RxNorm, and the Healthcare Common Procedure Coding System (HCPCS). Unstructured data in the clinical datasets were parsed into structured ones. Clinical data such as patient information, prescribed medications, and surgical information were selected from the formal data features that were processed as above. The created clinical datasets were reclassified by cancer type and used to generate a temporary clinical cancer registry. This temporary clinical cancer registry was moved to a manual input platform (i.e., an electronic case record form (e-CRF)), and the necessary data were added to complete the final clinical cancer registry. The completed clinical cancer registry was transferred to a DW and merged with clinical datasets. These processes were carried our weekly. Functions for data retrieval and analysis were implemented in a web-based platform of the K-CBP.

### 2.4. Data Validation and Monitoring

To ensure that we created the correct system, we designed three validation procedures, and applied one or more of these procedures to each step of the dataset construction process (Figure 3).

The goal of this process was to verify a large amount of data in a short period of time. Clinical datasets were validated by checking whether the process was performed (VA), comparing the number of original data and extracted data (VB), and finally checking the contents of sample data (VC). VA examines each step of the extract, transform, and load (ETL) procedure, which generates a temporary dataset called “temp mart”. VB compares the temporary dataset with a preprocessed dataset with respect to the amount of the data by calculating some predetermined indexes, including the number of extracted items. In VC, the data manager validates the contents by directly sampling the data. During this process, any content with an error is modified by the input platform. Each validation process can be performed independently and can be applied at any ETL stage.

### 2.5. Data Merging

The collected and processed datasets were merged and stored in an integrated DW. Data merging was performed on datasets such as biological sample information, genomic data, and national cancer control initiative data, starting with clinical datasets. The first step was to apply the same terminology standard to datasets based on clinical datasets with data matching. Then, each dataset was matched on the basis of the primary key of the clinical datasets. The primary key was used to restore the patient ID of each dataset and generate an alternative patient key. To avoid patient ID duplication and to match datasets, encrypted personal data were decrypted. After merging datasets, personal data were re-encrypted and stored in a closed system. Patients with duplicate IDs were disambiguated on the basis of their IDs at the NCC registry. Data from the clinical registries, biological samples, and gene mutations were transferred to the integrated DW where they could be linked to one another on the basis of the alternative patient key. Patient IDs could not be restored for data from the NCCI and external cancer registries. Instead of linking data to the clinical cancer registries at the NCC Hospital, we calculated summary statistics for NCCI data and presented them at the K-CBP.

### 2.6. Construction of the K-CBP

The K-CBP was constructed as a web-based data retrieval and analysis platform to utilize merged data. The platform was designed to connect with an integrated DW to reflect changes instantly. Data that can be utilized through the platform are shown in Figure 4.

### 2.7. Legal and Regulatory Processes

The K-CBP data were provided in strict compliance with the user rights management and privacy policy. Because of legal regulation, only a few medical staff can use data from the K-CBP. Data distribution was permitted only after institutional review board approval. Data de-identification, processing, and provision were carried out through an honest broker to minimize the possibility of personal information leakage.

## 3. Results

### 3.1. Collection of Data

The cancer-related data held by the NCC included clinical, genomic, tumor biomarker, and cancer registration statistics data (Table 1). These data were split into subcategories according to data types, such as cancer sample type. Clinical cancer registries are in use for clinical and research purposes on specific cancer sites, such as prostate, lung, pancreatic, kidney, and ovarian cancers, and others are under construction for colon, liver, stomach, breast, and thyroid cancers. Additionally, the external registries will also expand the cancer type. These cancer registration statistics data comprise information such as diagnosis, death, surgery, drug prescription, Surveillance, Epidemiology, and End Results (SEER) summary stages, personal or family income, and healthcare institution included in hospice and palliative care.

### 3.2. Construction of Clinical Datasets and Clinical Cancer Registries

We generated clinical cancer registries for 10 major cancers. Unstructured data during the registry construction by cancer type averaged 24.3% compared to the whole features but decreased to 7.64% after structuring. Table 2 shows the features of structured, unstructured, and manual input data by cancer type. 

More information about these terms and features is shown in Table 3 and Appendix A. Detailed information on the established registries for prostate, lung, pancreatobiliary, kidney, and ovarian cancers can be explored on sites linked to URL and QR code shown in Appendix A.

### 3.3. Data Merging

Clinical data from 515,780 cancer patients, NGS data from 280 patients, metadata of blood samples from 32,760 subjects, and metadata of tissue samples from 17,813 subjects were merged on the basis of the alternative key. In addition, cancer registration statistics data generated from the cancer control activities conducted by the four branches of the NCCI and external clinical cancer registries from separate cancer societies were transferred to the K-CBP. However, these data were not directly linked to the NCC clinical registries and the K-CBP because of the lack of information needed for re-identification.

### 3.4. Construction of the K-CBP

The K-CBP consists of a pre-generated dataset area and a dataset combination area. Through this, the K-CBP provides detailed information on each feature of the dataset. The user can inquire about the desired information through the K-CBP and reconstruct this new information to create a personalized dataset. Furthermore, the K-CBP provides status information about the data held by the platform, as shown in Figure 5.

On the web platform of the K-CBP, a user can query for the clinical registry data, analyze the features of different datasets, and carry out studies such as tracking and observing subjects who overlap in a specific period and for specific conditions. The generated personal dataset can be extracted and utilized after approval of the administrator. The National Cancer Data Center website (NCDC, http://www.cancerdata.kr) provides a detailed description of data held by the K-CBP, as well as a description of and data on the features of the retrieved function (Figure 6).

## 4. Discussion

The DW introduced in this paper is an integration of medical data that can enrich the quality of research and offer high-quality data services in medical fields. It is regularly being updated through strict evaluation steps to assure data quality. The DW is also the foundation of precision medicine, as it allows researchers or healthcare experts to trace patients’ cumulative records over several years. 

Although the established K-CBP has the potential to be used in precision medicine and clinical research, some limitations should be discussed. Genomic data are currently being integrated through the K-CBP, but the amount of data is insufficient to provide confidence in the results. Data on lifestyle and environment from various survey or study results were not integrated through the K-CBP. We expect that these datasets can be merged into K-CBP on the basis of the alternative key. On this platform, there are only 10 kinds of clinical cancer registries for each cancer type, combined with the number of cancers to be generated. Therefore, there is a limitation for research on rare and intractable cancers.

The merging of the generated datasets is a critical issue to be solved. When the source of the data is one institution, it is possible to merge the data using the common identifier used in that institution. However, when the sources of the data are multiple institutions, it is necessary to select a common identifier that covers institutions; most common identifiers that satisfy the conditions are not free from the laws and regulations imposed on the medical data. Data on lifestyle and environment, which are essential for precision medicine, are usually limited in terms of incorporation into the cancer big data platform because of the different clinical data and sources. Legal regulations in Korea that pertain to permission to access data, including personal information, should be eased. However, the legislation that concerns the authority to access these data is under discussion, and specific legal boundaries against the risk of using personal information should be proposed. 

Despite these limitations, the K-CBP attempts to establish and implement basic datasets for cancer-related clinical studies and precision medicine. It creates and manages high-quality datasets. Particularly, we tried to secure the quality of data through the structuring of unstructured data, creation of datasets, and use of a validation system for the ETL process. However, there are limitations on the validation of source data and the presence of features that cannot be structured, which should be addressed in the future. Unstructured data that cannot be structured depend on manual input, and there is a possibility of error. Validation of manual input data is only through direct human validation; it is not systematic in comparison to the validation of other datasets. Moreover, a large dataset takes a long time to manually input, which makes it difficult to immediately update the data. In addition, unstructured text-formatted data are generated as a dataset without processing, which makes it necessary for the researcher to perform secondary processing. To secure the quality of the data, it is necessary to devise a method of generating a dataset by structuring the manual input data in addition to the conventional method of shaping. Furthermore, it is necessary to reduce human error and ensure quality by constructing a system that can instantly reflect all the changes as the source data changes. 

## 5. Conclusions

In this study, we suggest the need for an integrated medical DW and introduce the construction process and expected outcomes from the utilization. K-CBP collects, links, and refines clinical, genomic, biological, statistical, and multi-institutional medical data under the jurisdiction of the law. Through this, comprehensive medical data for one subject (patient) are obtained. We developed a web-based platform that can utilize the collected data to enhance accessibility. Security is also taken into consideration through de-identification and user authority management over several stages. In the future, based on the expansion of features, data accumulation, and quality improvement, we intend to contribute to the improvement of the quality of public health by providing a system that can produce and enable healthcare experts to utilize high-quality base materials necessary for precision medicine.

## Figures and Tables

**Figure 1 ijerph-16-02290-f001:**
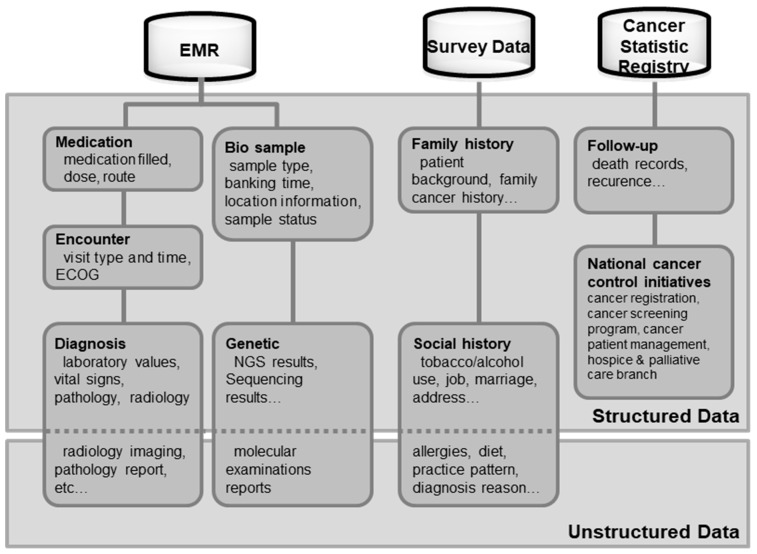
Overview of the integrated Korea Cancer Big Data Platform (K-CBP) by data type. ECOG, Eastern Cooperative Oncology Group performance status; EMR, electronic medical record; NGS, next-generation sequencing.

**Figure 2 ijerph-16-02290-f002:**
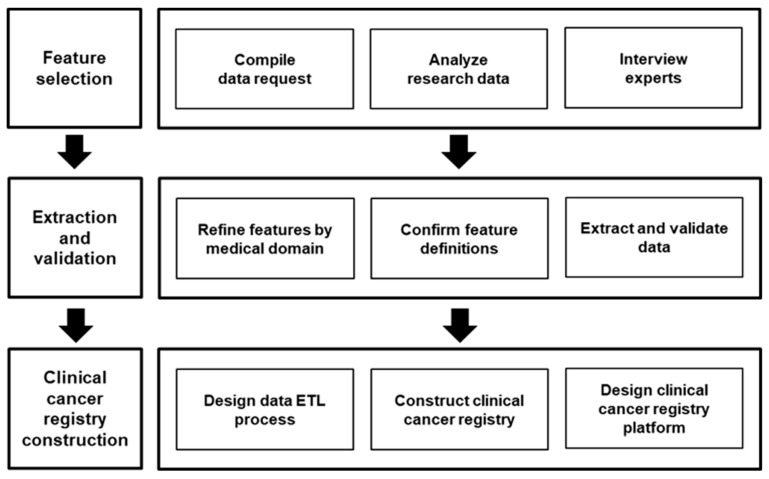
Clinical cancer registry construction and utilization process. ETL, extract, transform, and load.

**Figure 3 ijerph-16-02290-f003:**
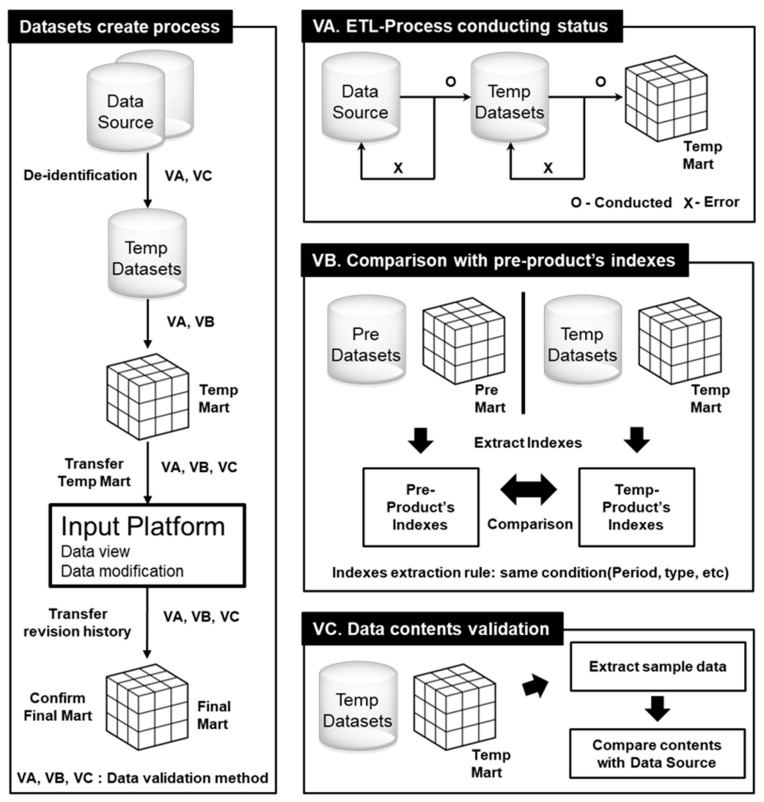
Data validation process. Temp, temporary.

**Figure 4 ijerph-16-02290-f004:**
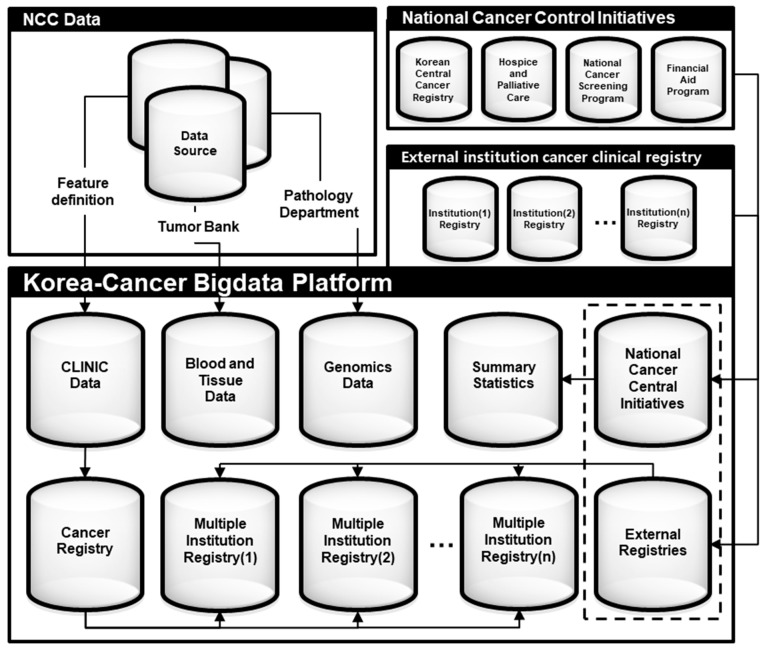
Dataset configuration of the Korea Cancer Big Data Platform (K-CBP). NCC, National Cancer Center.

**Figure 5 ijerph-16-02290-f005:**
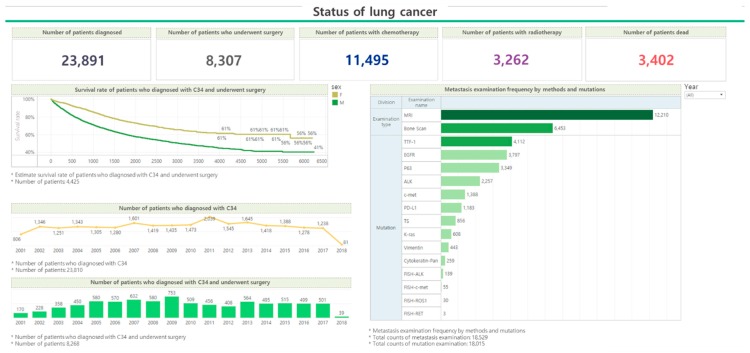
A screenshot of the Korea Cancer Big Data Platform (K-CBP) clinical cancer registry website showing the summary statistics for lung cancer as of December 2018.

**Figure 6 ijerph-16-02290-f006:**
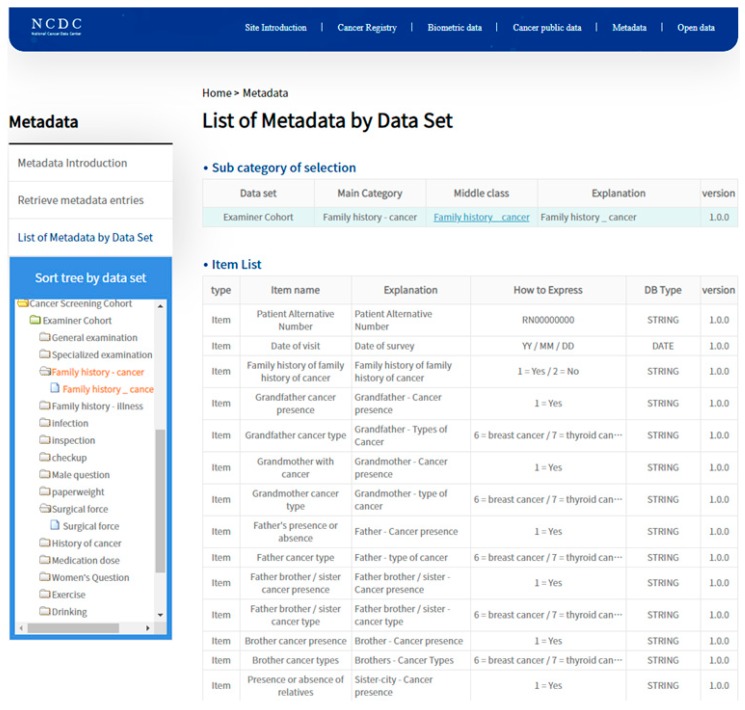
A screenshot of the National Cancer Data Center (NCDC) webpage displaying a list of items and a description for ovarian cancer (https://www.cancerdata.kr/dataSetMetaLst.do).

**Table 1 ijerph-16-02290-t001:** Number and type of items integrated in the Korea Cancer Big Data Platform (K-CBP).

Category	Subcategory	Number of Subjects	Description of Features								
Number of patients		515,780									
Medical records	Pathology records	925,599	Specimen type, method of examination, clinical and pathological diagnosis								
	Order sheet	11,703,931	Orders related to treatment and discharge								
Tumor bank	Blood sample	32,760	Information on pathologic stages; normal and tumor tissues, sample status, and location information								
	Tissue sample	17,813									
Genomics	NGS test	280	Mutations detected on panel-based NGS tests								
Clinical cancer registries at the NCC Hospital	Cancer – core features	144,944	Patient information, diagnosis, images, normal and tumor tissues, surgery, pathological and clinical stages, chemotherapy, radiation therapy, recurrence and metastasis, death								
	Prostate cancer	5167									
	Lung cancer	24,504									
	Pancreatobiliary cancer	9966									
	Kidney cancer	2886									
	Ovarian cancer	4240									
	Colorectal cancer	17,328									
	Liver cancer	12,932									
	Breast cancer	18,287									
	Gastric cancer	15,056									
	Thyroid cancer	10,404									
Cancer statistical registry data from the National Cancer Control initiatives	Korea Central Cancer Registry	2,745,050	Nationwide data on the diagnosis and treatment of cancer and survival of patients								
	National Cancer Screening Program	90,197,402	Data obtained from nationwide screening for stomach, liver, colorectal, breast, and uterine cervix cancers								
	Financial Aid Program for Cancer Patients	543,325	Data relevant to financial aid for low-income cancer patients								
	Hospice and Palliative Care	56,433	Information on performance status (ECOG), admission to and discharge from hospice institutions, and the use of hospice care								
Clinical cancer registries from external sources	Prostate cancer	7934	Complications, surgery								
	Lung cancer	3496	Results of biopsy, gene mutation, surgery								
	Pancreatic cancer	538	Tumor, physical examination findings, surgery								

ECOG, Eastern Cooperative Oncology Group; NGS, next-generation sequencing; NCC, National Cancer Center.

**Table 2 ijerph-16-02290-t002:** Features of structured, unstructured, and manual input data in the clinical cancer registry by cancer type.

Cancer Type	Data Type			Total
	Structured	Unstructured	Manual Input	
Prostate cancer	165	66	13	244
Lung cancer	146	85	4	235
Pancreatobiliary cancer	319	34	54	407
Kidney cancer	369	70	41	480
Ovarian cancer	428	59	32	519
Colorectal cancer	230	51	84	365
Liver cancer	216	84	50	350
Breast cancer	228	85	27	340
Gastric cancer	175	141	14	330
Thyroid cancer	156	244	4	404

Structured data, data that can be represented by a specific number or word and whose format is roughly defined; unstructured data, free-text format data; manual input data, data that cannot be automatically entered through computerization.

**Table 3 ijerph-16-02290-t003:** Definition of terms.

Term	Definition
Alternative patient key	A primary key that replaces a direct identifier with a random 8-digit number
De-identification	Elimination of direct identifiers and quasi-identifiers so that individuals cannot be identified
Clinical cancer registry	Outcome data such as diagnosis, treatment, and surgery that are selected among cancer clinical data; dataset refined in a form that can be used meaningfully
National cancer control initiative data	Cancer-related data collected under a nationally led project
External clinical cancer registry	Cancer-related clinical data from multiple institutions, including diagnosis, treatment, or surgery; dataset selected and refined for outcome data
Structured data	Data that can be represented by a specific number or word and whose format is standardized
Unstructured data	Free-text format data
Manual input data	Data that cannot be automatically imported through computerization

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
