# Peer review of "The Korea Cancer Big Data Platform (K-CBP) for Cancer Research"

_ijerph, 2019, doi:10.3390/ijerph16132290_

Round 1
Reviewer 1 Report
This paper reflects a huge effort on the part of the researchers and others involved. Congratulations for pulling this off!
Main points:
The authors brought up the issue of legal and regulatory challenges in the discussion and abstract, but I didn’t see any further explanations of those changes or challenges in the text. Please elaborate on what you recommend and provide a bit of insight (if you can) from doing this project.
Minor edits:
There are a few minor grammar edits that I think should be made. I was not able to catch all of them but here are a few corrected in underlined text and strikethroughs.
Why do the references have two closing brackets? Remove and replace with only 1 bracket.
Abstract – see underlined additions to text
Abstract: Data warehousing became the most important technology to address recent advances in precision medicine. However, a generic clinical data warehouse does not address unstructured and insufficient data. In precision medicine, it is essential to develop a platform that can collect and utilize these data. Data were collected from electronic medical records, genomic sequences, tumor biopsy specimens, and national cancer-control initiative databases in the National Cancer Center, Korea. Data were de-identified and stored in a safe, independent space. Unstructured clinical data were standardized and incorporated into cancer registries and linked to cancer genome sequences and tumor biopsy specimens. Finally, national cancer-control initiative data from the public domain were independently organized and linked to cancer registries. We have constructed a system for collecting and providing various cancer data called the Korea-Cancer Bigdata Platform(K-CBP). The K-CBP could be used for cancer research, but legal and regulatory aspects of data distribution and usage need to be addressed. Nonetheless, the system will continue to collect cancer-related resources and hopefully facilitate precision-based research.
Introduction:
Clinical oncology is experiencing a major paradigm shift with the advent and dissemination of high-throughput genome sequencing technologies and molecularly targeted therapies.
Please choose a different word for the following sentence: Initial versions of clinical DWs simply stored raw data that were generated in the course of diagnosis and treatment without additional purification editing?
Edit: In addition, a web-based platform has been developed based on the registry constructed to actively utilize clinical data
Edit: the need for a platform with a large scale and range has arisen, and a big data platform instead of a simple web-based platform has been developed
Edit: In this paper, we introduce the Korea-Cancer Bigdata Platform (K-CBP), a multi-database framework that collects clinical, genomic, imaging, and biobank data using a secure de-identification technology and allows for clinical research and practice future research.
Methods
I am a bit confused by the structure of Figure 1 compared to the way the data sources section was written. Can you write the data sources section to match what is in the figure?
Results
Table 1 – why is the N of ~2018.08 a decimal? Shouldn’t this be an integer? And what is the N? The total number of variables included in the database? Are the rest of the N’s patient records or records of individual events, tests, and diagnoses?
Table 2 – insert a superscript that explains what each of the data type terms means (structured, unstructured, manual input).
Get rid of Figure 5. It is not needed and I am worried about patient PHI being shown in the table.
I feel like the data merging section should be in the Methods section.
Rename the “Providing data” section “Legal and regulatory processes” and move to Methods.
Discussion
You mentioned that environmental and lifestyle data are not available. If there are plans to include this type of data, please indicate so and list potential sources if possible.
Edit: … and has created and managed high-quality
Edit: we expect to be able to implement a complete data platform that can be utilized in precision medicine.
Author Response
Responses to the reviewers’ comments(1st)
First, thank you for your comments. After checking the comments, we carefully improved our paper. The details are as follows;
Reviewer # 1
Main points
The authors brought up the issue of legal and regulatory challenges in the discussion and abstract, but I didn’t see any further explanations of those changes or challenges in the text. Please elaborate on what you recommend and provide a bit of insight (if you can) from doing this project.
Answer: Under current law, the collection and use of medical data are quite limited. For the broader use of medical data, the relaxation of legal regulations should be accompanied by gradual progress, and we have described this in our discussion.
Minor edits
There are a few minor grammar edits that I think should be made. I was not able to catch all of them but here are a few corrected in underlined text and strikethroughs.
Why do the references have two closing brackets? Remove and replace with only 1 bracket.
Answer: We have modified Reference’s bracket.
Abstract:
See underlined additions to text … However, a generic clinical data warehouse does not address unstructured and insufficient data. In precision medicine, it is essential to develop a platform that can collect and utilize these data. Data were collected from electronic medical records, genomic sequences, tumor biopsy specimens, and national cancer-control initiative databases in the National Cancer Center, Korea. … We have constructed a system for collecting and providing various cancer data called the Korea-Cancer Bigdata Platform(K-CBP). … Nonetheless, the system will continue to collect cancer-related resources and hopefully facilitate precision-based research.
Answer: We modified underlined words (page 2).
Introduction:
Clinical oncology is experiencing a major paradigm shift with the advent and dissemination of high-throughput genome sequencing technologies and molecularly targeted therapies.
Answer: We have modified the expressed words (page 3).
Please choose a different word for the following sentence: Initial versions of clinical DWs simply stored raw data that were generated in the course of diagnosis and treatment without additional purification editing?
Answer: We have modified the expressed words (page 3).
Edit: In addition, a web-based platform has been developed based on the registry constructed to actively utilize clinical data
Answer: We have modified the expressed words (page 3).
Edit: the need for a platform with a large scale and range has arisen, and a big data platform instead of a simple web-based platform has been developed
Answer: We have modified underlined words (page 4).
Edit: In this paper, we introduce the Korea-Cancer Bigdata Platform (K-CBP), a multi-database framework that collects clinical, genomic, imaging, and biobank data using a secure de-identification technology and allows for clinical research and practice future research.
Answer: We have modified underlined words (page 4).
Methods:
I am a bit confused by the structure of Figure 1 compared to the way the data sources section was written. Can you write the data sources section to match what is in the figure?
Answer: We have modified data sources contents to match in figure 1. (page 5)
Results:
Table 1 ? why is the N of ~2018.08 a decimal? Shouldn’t this be an integer? And what is the N? The total number of variables included in the database? Are the rest of the N’s patient records or records of individual events, tests, and diagnoses?
Answer: We have replaced ‘N’ with ‘Number of subject’ (table1).
Table 2 ? insert a superscript that explains what each of the data type terms means (structured, unstructured, manual input).
Answer: We have added superscript of each data type terms means (table2).
Get rid of Figure 5. It is not needed and I am worried about patient PHI being shown in the table.
Answer: We have deleted figure 5 (page 20).
I feel like the data merging section should be in the Methods section.
Answer: This section’s contents show the result of data merging, so we modified the section’s contents to fit RESULT paragraph. (page 20)
Rename the “Providing data” section “Legal and regulatory processes” and move to Methods.
Answer: We have modified and moved this section to the method from the result. (page 12)
Discussion
You mentioned that environmental and lifestyle data are not available. If there are plans to include this type of data, please indicate so and list potential sources if possible.
Answer: We have added example and merging method of these data. (page 26)
Edit: … and has created and managed high-quality
Answer: We have modified underlined words (page 26).
Edit: we expect to be able to implement a complete data platform that can be utilized in precision medicine.
Answer: We have modified underlined words (page 27).
Reviewer 2 Report
This paper describes the construction of data warehouse (DW) for use in cancer research in Korea, using and linking databases from private and public sources. The paper contains impressive quantities of material and knowledge.
However a number of issues should be resolved to improve understanding of the principles of the construction and functioning of this DW.
General comments
Using the term ‘cancer registry’ for a cancer site-specific dataset within the warehouse may be inappropriate. A cancer registry is a systematic collection of data about cancer, where data collection is a continuous process, informed by quality control and feedback. However, the database called ‘cancer registry’ within the described warehouse seems to be constructed periodically, without a possibility to control the data quality due to de-identification and without the continuous data flow. It would be better to replace ‘cancer registry’ with another term, e.g. ‘cancer-specific dataset’.
Another possibly incorrectly used term is ‘public domain’ or ‘public data’. It is said that public domain data were generated from four national cancer-control activities, including the Korea Central Cancer Registry (KCCR). However on the KCCR web-site there is no access to ‘public domain data’ apart from aggregated statistics, which are not useful for linking patients’ data. Do the authors mean ‘population-based’ or ‘aggregated’ when they say ‘public’? If the data were ‘aggregated’, how could they have been linked to the other patients’ records in the DW? In any case, this needs to be clarified throughout the manuscript.
The term ‘collection’ or ‘collected’ in K-CBP is used with respect to data. However, from the description of the procedures it appears that data are extracted from other databases and then (processed and) entered into the K-CBP. Therefore, it would be more appropriate to use terms ‘inclusion’, ‘integration’, etc. instead of ‘collection’.
Although the text is written in good English, the text should be reviewed carefully, as occasional sentences do not make sense. For example, in abstract, third sentence is missing a subject and the fourth sentence is missing a verb; also there are two freely standing words at the end of Introduction, etc.
Finally, the language is often approximate, imprecise or confusing and should be revised; some hints are provided below.
The citations in the text are enclosed in asymmetric brackets, eg [1]]. Why not use just simple brackets, eg [1]?
Specific comments
Abstract
It is said that “the K-CBP could be used for cancer research, but legal and regulatory aspects of data distribution and usage need to be addressed”, but should these have been addressed before constructing this warehouse? Does it mean that the data in the K-CBP system could only be used at present by its engineers? A statement should address this aspect.
Introduction
Replace ‘molecularly’ with ‘molecular’.
In the statement “Now that this precision-based approach has become part of standard care in clinical oncology” it should be specified that this relates to high income countries.
Reformulate “databases have attempted to”.
Consider shortening this section.
Material and Methods
Data sources
Data were obtained ‘about’ patients and not ‘from’ patients.
Information ‘was’ collected and not ‘were’ collected.
‘normal or cancer tissues’ would be better as ‘tissues’
The sentence “All data collected on the platform were de-identified and stored in an independent DW.”:
As the data come from other databases it might be more appropriate to change to ‘included in’ or ‘imported in’ instead of ‘collected on’;
The ‘independent DW’ suggests that the process is not a continuous one, but rather it is executed in batches and each new version of the DW would require the same data extraction from various databases. The incremental nature of the DW should be made explicit, as well as dealing with previous versions of a DW: would they be discarded or compared with the new version to consolidate any changes?
De-identification of datasets
In the above title the ‘datasets’ should be replaced by ‘records’, as datasets do not present identification risks.
It would be useful to list the items that are considered to be ‘direct identifiers’ and all those that are considered (in combination) the ‘quasi identifiers’.
It is not clear if the alternative keys identified each individual uniquely. Could this be specified? If so, it would be interesting to explain how this was done.
It is said that the quasi identifiers were deleted, but potentially there are many of these and their deletion may limit the usefulness of the resulting datasets. Could this be explained?
Presumably, the ‘closed system’ stores the link between the true ID and the alternative key. Who (and how many people) has access to this ‘closed system’ and who has access to the resulting database?
If the alternative keys are changed every time to disable tracking, does it mean that all data quality issues need to be resolved before de-identification? How is follow-up or update of information dealt with?
Creation of clinical datasets and cancer registries
With reference to point 1, ‘cancer registries’ should be replaced by ‘datasets’ or other more appropriate term in the above title and throughout the manuscript.
The personal ‘lifelog’ data should be explained.
With reference to point 15.b, it appears that the clinical cancer registries (which should be renamed to datasets or other similar term – see point 1), have to be re-created each time a batch update of the DW is conducted. This should be specified here or discussed in Discussion.
It also appears that there is no possibility for improving data quality once this dataset is created. This should also be specified here or in Discussion.
Page 4, 1st paragraph, 3rd line from bottom: ‘managed’ should be replaced by ‘merged’.
Data validation and monitoring
It is not specified if and how found errors are corrected. If an error pertains to the input data, are they traced back to the original data source?
Data merging
Recourse to the patient’s ID is made during merging. While this is understandable, should the encryption be done at this stage rather than in a previous stage described in section ‘De-identification…’?
Missing parts
The roles and access rights should be explained. For example, the administrator is mentioned on page 13 and an ‘honest broker’ on page 15, but it is not explained who they are.
The possibilities and modalities of access to data from people/researchers external to the DW should be described.
If the K-CBP is accessible from outside, the website address should be provided. If it is not accessible, it should be specified.
Results
Results contain a lot of information that belongs to Methods. In Results it might be useful also to provide the numbers of each type of institutions that provide data and the frequency of the provision. It can also report the results of the validation checks of the data.
Collection of data
In this paragraph one learns that the collected data were updated weekly. This should have been mentioned in Methods, along with a description of the update process. It should be explained in Methods.
Construction of clinical datasets and clinical cancer registries
Use of term ‘registry’ should be revised with reference to point 1.
Data merging
Data ‘from’ patients should be replaced with data ‘on’ patients.
The fact that the data from national sources were de-indentified prior to their import in the DW should be (also) mentioned in Methods.
Construction of the K-CBP
Data ‘from’ patients should be replaced with data ‘on’ patients.
If the data for NCDC, represented in Figure 7, are integrated separately into the K-CBP, what is the added value of this large DW over having the NCDC data separately located within NCDC structure? Isn’t their integration into K-CBP just adding an extra obstacle on the way to the data?
Providing data
This paragraph belongs to Methods section. However, more details should be provided on the categories of users that are eligible for (applying to) using these data. Are there any application forms? If not, what is the address of the website? Are the approvals of Institutional Review Boards of any institution (domestic and foreign) accepted?
The Results section could provide the number of users or studies that were supported by the K-CBP to date since its inception.
Discussion
It would be preferable to start with describing all advantages of the constructed DW and its potential uses (they should be added), before escribing the limitations.
Each described limitation should be accompanied by a suggestion of how it could be overcome.
Are there other similar DW which could provide a good comparison to the presented work?
Please refer to point 3 for revision of terminology used in the first paragraph.
Conclusion
Conclusion is not really conclusion, but continuation of Discussion. The whole paragraph should be moved to Discussion section.
Conclusion should summarize briefly the findings (1 sentence), explain what do they mean and possibly outline the plans for future.
Tables
Table 1
As in DW the content might be more important that the source, the table might be better arranged as showing the content in the first column, the numbers of cases in the second, the source of data in the third and explanatory notes in the fourth.
The column headers should be shown on each page.
The layout of this table should be improved, so that it is clear which description applies to which category.
It should be specified if all records that are shown in this table are linked mutually. For example, prostate cancer registry is mentioned in the table twice, which suggests that there are two sources of data for prostate cancer. What are the differences between them and what are the consequences for use?
Line ‘Tumour bank’:
‘pathology, stage,’ should replace ‘pathologic stages;’
‘normal’ should be avoided in connection with ‘tissue’ – see point 14.
Mention of tissue should be aligned with ‘Tissue sample’ line.
How are the blood samples and tissue samples stored in the DW? Are they physically stored there or just the data that were derived from those samples? This should be made transparent.
Line ‘Clinical cancer registries…’
Word ‘core’ would be better than ‘common’.
It should be made clear that the text in the first and last column is relevant to all lines in this cell.
Line ‘Public domain data…’
‘Public domain’ should be resolved with reference to point 2.
Table 2
‘items’ might favourably replace ‘data’ in the title of this table.
Table 3
It does not sound right to define ‘alternative key’ as ‘primary key’.
The difference between ‘clinical cancer registry’ and ‘external clinical cancer registry’ is unclear and should be clarified. They could also be listed next to each other for easier visual comparison of the differences. Examples would be useful.
Structured data: ‘roughly defined’ might be better as ‘standardised’.
Manual input data:
It would be useful to give example of these data and explain how they are input (workforce requirement, sources, frequency,…).
‘entered’ might be better as ‘imported’.
Consider if other terms should be defined, based on the other comments.
Figures
Figure 1
From the content of the boxes linked to ‘Public Data’, these are unlikely to be ‘Public’ - not everybody can access them. See also point 2 for further comments on suggested improvement.
It might be useful to add ‘location’ in the box ‘Bio sample’.
The indenting should be identical in all boxes.
‘lab values’ should be shown in full, ‘laboratory values’.
‘recur’ should be replaced by a proper word.
Figure 2
The title should be modified in accordance with comments 1 and 22.
Figure 3
It would be helpful if an example could be provided for each of the data validation method VA, VB, VC.
Figure 4
Terminology should be revised for all resulting databases called ‘registries’, in accordance with point 1.
It should be explained what is meant by ‘External registries’.
If ‘CLINIC’ is not an abbreviation, it should not be shown with all capital letters. If it is an abbreviation, it should be explained.
Figures 5, 6 and 7
Apart from the figures containing Korean characters, the screen shots do not have a sufficient resolution for reader to understand what is shown. The resolution of the screenshots must be improved.
Supplementary material
Tables
The Excel workbook should be formatted in such a way that each table be printed automatically on a single page (now each worksheet is spread on three pages with no row headers apparent on 2nd and 3rd page).
The text in the tables should be reviewed carefully for errors and abbreviations. All abbreviations and shorthand expressions should be explained in footnotes.
The number of each table should be included in the Table title, so that it corresponds to the list on page 16 of the manuscript.
Suggested template for the title of these tables: ‘Contents of clinical dataset: prostate cancer’
Material 2
Along with the URL, there should be some guidance on what one finds on a given URL and how this source could be used.
Table S11
This table, listed in manuscript on page 16, seems to be missing from the submitted material available for review.
Author Response
Responses to the reviewers’ comments(1st)
First, thank you for your comments. After checking the comments, we carefully improved our paper.The details are as follows;
Reviewer # 2
General comments
Using the term ‘cancer registry’ for a cancer site-specific dataset within the warehouse may be inappropriate. A cancer registry is a systematic collection of data about cancer, where data collection is a continuous process, informed by quality control and feedback. However, the database called ‘cancer registry’ within the described warehouse seems to be constructed periodically, without a possibility to control the data quality due to de-identification and without the continuous data flow. It would be better to replace ‘cancer registry’ with another term, e.g. ‘cancer-specific dataset.’
Answer: Our datasets can control the data quality through three steps: rigorous construction processes, the input of corrections, and periodic updates. All history of processes during steps have logged in order to explore cumulative data sets conditionally. However, our datasets different from the existing cancer registry. For those reasons, we named our datasets ‘clinical cancer registry’ instead ‘cancer registry’.
Another possibly incorrectly used term is ‘public domain’ or ‘public data.’ It is said that public domain data were generated from four national cancer-control activities, including the Korea Central Cancer Registry (KCCR). However, on the KCCR web-site there is no access to ‘public domain data’ apart from aggregated statistics, which are not useful for linking patients’ data. Do the authors mean ‘population-based’ or ‘aggregated’ when they say ‘public’? If the data were ‘aggregated’, how could they have been linked to the other patients’ records in the DW? In any case, this needs to be clarified throughout the manuscript.
Answer: In this study, we used the term ‘public data’ meaning ‘statistical data related to national’ rather than ‘open data.’ Because of legal regulations, although we collected national data such as cancer registry and cancer screening data, we can only provide meta information like aggregated statistics. However, since the meaning of this word can be confusing, we switched ‘public data’ to ‘cancer registration statistics data.’
The term ‘collection’ or ‘collected’ in K-CBP is used with respect to data. However, from the description of the procedures it appears that data are extracted from other databases and then (processed and) entered into the K-CBP. Therefore, it would be more appropriate to use terms ‘inclusion’, ‘integration’, etc. instead of ‘collection’.
Answer: K-CBP collect and merge a variety of cancer datasets. So we used the term ‘collect’ until ‘merge section’ and modified ‘collect’ to ‘integrate’ from ‘merge section’.
Although the text is written in good English, the text should be reviewed carefully, as occasional sentences do not make sense. For example, in abstract, third sentence is missing a subject and the fourth sentence is missing a verb; also there are two freely standing words at the end of Introduction, etc.Finally, the language is often approximate, imprecise or confusing and should be revised; some hints are provided below.
Answer: Grammar, vocabulary, and sentences will receive professional revise while reviewing. However, we will send you the pre-revised version at this time because it spends a few days. We will revise it as soon as possible.
The citations in the text are enclosed in asymmetric brackets, e.g. [1]]. Why not use just simple brackets, e.g. [1]?
Answer: We have changed the double bracket to a single bracket.
Specific comments
Abstract
It is said that “the K-CBP could be used for cancer research, but legal and regulatory aspects of data distribution and usage need to be addressed”, but should these have been addressed before constructing this warehouse? Does it mean that the data in the K-CBP system could only be used at present by its engineers? A statement should address this aspect.
Answer: Korean laws in terms of medical data strictly restrict the authorization for accessing patients’ personal information; only medical staffs and some experts who manage the data, therefore, can access the data with apparent reasons. For instance, data can be used for research and statistical calculation, and the data of K-CBP that is currently constructed is utilized by these people.
Introduction
Replace ‘molecularly’ with ‘molecular’.
Answer: We modified molecularly’ to ‘molecular’. (page 3).
In the statement “Now that this precision-based approach has become part of standard care in clinical oncology,” it should be specified that this relates to high-income countries.
Answer: We added it relates to high-income countries (page 3)
Reformulate “databases have attempted to”.
Answer: We modified the sentence on page 3
Consider shortening this section.
Answer: Introduction section states the rationales and the purpose of this study. A data warehouse that we are introducing may be unfamiliar to worldwide as well as domestic readers since the DW has rarely been introduced. We also want to share how it is developing as well as show how personal information is protected with techniques that we introduced, which presents the novelty of this paper. After discussions with authors, we concluded that all statements and paragraphs in the introduction should remain.
Material and Methods
Data sources
Data were obtained ‘about’ patients and not ‘from’ patients.
Answer: We added additional data sources, not patients. (page 5)
Information ‘was’ collected, and not ‘were’ collected.
Answer: We modified ‘were’ to ‘was.’ (page 5)
‘normal or cancer tissues’ would be better as ‘tissues.’
Answer: We modified ‘normal or cancer tissues’ to ‘tissues.’ (page5)
The sentence “All data collected on the platform were de-identified and stored in an independent DW.”: As the data come from other databases it might be more appropriate to change to ‘included in’ or ‘imported in’ instead of ‘collected on’
Answer: We modified ‘collected on’ to ‘imported in’ (page 5)
The sentence “All data collected on the platform were de-identified and stored in an independent DW.” The ‘independent DW’ suggests that the process is not a continuous one, but rather it is executed in batches and each new version of the DW would require the same data extraction from various databases. The incremental nature of the DW should be made explicit, as well as dealing with previous versions of a DW: would they be discarded or compared with the new version to consolidate any changes?
Answer: The process of building a data warehouse can be divided into two steps: temporary generation step and confirm step. Every new DW is compared with prior DW in the temporary generation step. (Figure 3, VB) All processes of the DW generation are logged, which allows us to compare a new version of DW with previously generated DWs each other in order to confirm improvements as well as determine the nature of it.
We additionally explained about logging the process of the generation on page 5.
De-identification of datasets
In the above title the ‘datasets’ should be replaced by ‘records’, as datasets do not present identification risks.
Answer: We replaced ‘datasets’ to ‘records’. (page 6)
It would be useful to list the items that are considered to be ‘direct identifiers’ and all those that are considered (in combination) the ‘quasi identifiers’.
Answer: We added a sample of direct identifiers. (page 6)
It is not clear if the alternative keys identified each individual uniquely. Could this be specified? If so, it would be interesting to explain how this was done.
Answer: The alternative patient key was made using social security number which is the unique identification number assigned to every legal citizen in Korea. We added this explanation on page 7, the second paragraph.
It is said that the quasi identifiers were deleted, but potentially there are many of these and their deletion may limit the usefulness of the resulting datasets. Could this be explained?
Answer: The level of data deletion was determined depending on the level of user’s authority. Purpose of use or information provider’s agreement, the user has different access authorities. It evaluates whether provided data are useful or not. Related contents were added on page 7.
Presumably, the ‘closed system’ stores the link between the true ID and the alternative key. Who (and how many people) has access to this ‘closed system’ and who has access to the resulting database?
Answer: The only honest brokers who are responsible for providing data can access to this ‘closed system.’ We added these contents on page 7.
If the alternative keys are changed every time to disable tracking, does it mean that all data quality issues need to be resolved before de-identification? How is follow-up or update of information dealt with?
Answer: DW’s contents were checked by medical staff who has authority. Medical staff combines multiple quasi-identifiers to identify individuals and compare it with data sources.
Creation of clinical datasets and cancer registries
With reference to point 1, ‘cancer registries’ should be replaced by ‘datasets’ or other more appropriate term in the above title and throughout the manuscript.
Answer: As mentioned above, the datasets we built are also a kind of registry, so we will continue to use the term ‘clinical cancer registry’.
The personal ‘lifelog’ data should be explained.
Answer: We explained personal lifelog data on page 7.
With reference to point 15.b, it appears that the clinical cancer registries (which should be renamed to datasets or another similar term? see point 1), have to be re-created each time a batch update of the DW is conducted. This should be specified here or discussed in Discussion.
Answer: We added the creation period on page 8.
It also appears that there is no possibility for improving data quality once this dataset is created. This should also be specified here or in Discussion.
Answer: We added a discussion about your comment in Discussion. New data is weekly uploaded on the system and merged into existing datasets. During the process, rigorous verification is applied to evaluate the quality of data and any data with an error would be modified by the input platform. In addition, data managers evaluate the system through logged history to improve the quality of data. Thus, the system has multiple steps to assure and improve the quality of whole dataset.
Page 4, 1st paragraph, 3rd line from bottom: ‘managed’ should be replaced by ‘merged’.
Answer: We replaced ‘managed’ with ‘merged’. (page 8)
Data validation and monitoring
It is not specified if and how found errors are corrected. If an error pertains to the input data, are they traced back to the original data source?
Answer: Error correction is done at data contents validation step by input platform. (Figure 3) We added related contents on page 8.
Data merging
Recourse to the patient’s ID is made during merging. While this is understandable, should the encryption be done at this stage rather than in a previous stage described in section ‘De-identification…’?
Answer: To avoid patient’s ID duplicate and to match datasets, encrypted personal data were decrypted. After merging datasets, personal data were re-encrypted and stored in a closed system. We added it on page 10.
Missing parts
The roles and access rights should be explained. For example, the administrator is mentioned on page 13 and an ‘honest broker’ on page 15, but it is not explained who they are.
Answer: We have added the explanation of an honest broker on page 7.
The possibilities and modalities of access to data from people/researchers external to the DW should be described.
Answer: Our data can access who has authority. because of legal regulation, only a few medical staff can use our data. We added these contents on page 12, the legal and regulatory processes section.
If the K-CBP is accessible from outside, the website address should be provided. If it is not accessible, it should be specified.
Answer: We have provided K-CBP website address on page 27 as below:
The National Cancer Data Center website (NCDC, http://www.cancerdata.kr) provide detailed description of data held by the K-CBP, as well as features of retrieve function and description
Results
Results contain a lot of information that belongs to Methods. In Results it might be useful also to provide the numbers of each type of institutions that provide data and the frequency of the provision. It can also report the results of the validation checks of the data.
Answer: Due to the character of K-CBP data, it is difficult to visualize the results of validation checks because the most important validation step depends on people. All data sets, except for the multi-institution(external) clinical cancer registries and cancer statistical registries, are comprised of data from national cancer center Korea alone, and most have a one-week update cycle.
Collection of data
In this paragraph one learns that the collected data were updated weekly. This should have been mentioned in Methods, along with a description of the update process. It should be explained in Methods.
Answer: We deleted the contents should be explained in Methods. (page 13)
Creation of clinical datasets and clinical cancer registries
Use of term ‘registry’ should be revised with reference to point 1.
Answer: As mentioned above, the datasets we built are also a kind of registry, so we will continue to use the term ‘clinical cancer registry.’
Data merging
Data ‘from’ patients should be replaced with data ‘on’ patients.
Answer: We modified this paragraph suitable for the results section. (page 20)
The fact that the data from national sources were de-identified before their import in the DW should be (also) mentioned in Methods.
Answer: We mentioned this fact in the method section. (de-identified paragraph, page 6)
Construction of the K-CBP
Data ‘from’ patients should be replaced with data ‘on’ patients.
Answer: We switched ‘from’ to ‘on’
If the data for NCDC, represented in Figure 7, are integrated separately into the K-CBP, what is the added value of this large DW over having the NCDC data separately located within NCDC structure? Isn’t their integration into K-CBP just adding an extra obstacle on the way to the data?
Answer: NCDC shows K-CBP’s summary and metadata. In addition, it shows the construction method of cancer datasets, feature information and retrieves function of cancer dataset’s feature list.
Providing data
This paragraph belongs to Methods section. However, more details should be provided on the categories of users that are eligible for (applying to) using these data. Are there any application forms? If not, what is the address of the website? Are the approvals of Institutional Review Boards of any institution (domestic and foreign) accepted?
Answer: We replaced this paragraph in the method section. And we added ‘using data’ related contents.
The Results section could provide the number of users or studies that were supported by the K-CBP to date since its inception.
Answer: Since the construction of the K-CBP was completed recently, there is no example yet of utilizing the data of K-CBP.
Discussion
It would be preferable to start with describing all advantages of the constructed DW and its potential uses (they should be added), before escribing the limitations.
Answer: We added advantages and potential uses of DW. (page 26)
Each described limitation should be accompanied by a suggestion of how it could be overcome.
Answer: We added how to overcome these limitations on page 26.
Are there other similar DW which could provide a good comparison to the presented work?
Answer: There a few similar DW support simple and advanced retrieval to utilize diagnosis or medical research such as Dr. Warehouse. But they did not collect a variety of medical data except clinical data.
Conclusion
Conclusion is not really conclusion, but continuation of Discussion. The whole paragraph should be moved to Discussion section.
Answer: We have moved this paragraph in the Discussion section. (page 27)
Conclusion should summarize briefly the findings (1 sentence), explain what do they mean and possibly outline the plans for future.
Answer: We re-wrote the conclusion. (page 27)
Tables
Table 1
The column headers should be shown on each page.
Answer: We modified table headers.
It should be specified if all records that are shown in this table are linked mutually. For example, prostate cancer registry is mentioned in the table twice, which suggests that there are two sources of data for prostate cancer. What are the differences between them and what are the consequences for use?
Answer: The prostate cancer at the National Cancer Center Hospital collects data based on various ina single hospital and prostate cancer in Clinical Cancer Registiry from external sources collects data based on a number of academic hospital.
How are the blood samples and tissue samples stored in the DW? Are they physically stored there or just the data that were derived from those samples? This should be made transparent.
Answer: Blood samples and tissue samples are not sotred physically in the DW. Those are stored in the the tissue bank. In the DW, there is information derived from the those samples. We have addeded these description in the “Description of features” in Table 1.
Word ‘core’ would be better than ‘common’.
Answer: We modified ‘common’ to ‘core’
Table 2.
‘items’ might favourably replace ‘data’ in the title of this table.
Answer: We modified ‘data’ to ‘items’
Table 3.
The difference between ‘clinical cancer registry’ and ‘external clinical cancer registry’ is unclear and should be clarified. They could also be listed next to each other for easier visual comparison of the differences. Examples would be useful.
Answer: The prostate cancer at the National Cancer Center Hospital collects data based on various ina single hospital and prostate cancer in Clinical Cancer Registiry from external sources collects data based on a number of academic hospital.
Structured data: ‘roughly defined’ might be better as ‘standardized’.
Answer: We modified ‘roughly defined’ to ‘standardized’
‘entered’ might be better as ‘imported’.
Answer: We modified ‘entered’ to ‘imported’
Figures
Figure 1
From the content of the boxes linked to ‘Public Data’, these are unlikely to be ‘Public’ - not everybody can access them. See also point 2 for further comments on suggested improvement.
Answer: We modified ‘Public Data’ to ‘Cancer Statistic Registry’
It might be useful to add ‘location’ in the box ‘Bio sample’.
Answer: We added location
‘lab values’ should be shown in full, ‘laboratory values’.
Answer: We modified ‘lab values’ to ‘laboratory values’
‘recur’ should be replaced by a proper word.
Answer: We modified ‘recur’ to ‘recurrence’
Figure 4
Terminology should be revised for all resulting databases called ‘registries’, in accordance with point 1.
Answer: We changed figure
It should be explained what is meant by ‘External registries’.
Answer: We modified ‘External registries’ to ‘External institution cancer clinical registry’
If ‘CLINIC’ is not an abbreviation, it should not be shown with all capital letters. If it is an abbreviation, it should be explained.
Answer: We replaced ‘CLINIC’ with ‘Clinic’
Figures 5, 6 and 7
Apart from the figures containing Korean characters, the screen shots do not have a sufficient resolution for reader to understand what is shown. The resolution of the screenshots must be improved.
Answer: We changed screen shots high resolution version. And we also changed language Korean to English.
Supplementary material
Tables
The Excel workbook should be formatted in such a way that each table be printed automatically on a single page (now each worksheet is spread on three pages with no row headers apparent on 2nd and 3rd page).
Answer: We adjusted page spacing.
The number of each table should be included in the Table title, so that it corresponds to the list on page 16 of the manuscript.
Answer: We added table number refer to page 16 of the manuscript.
Suggested template for the title of these tables: ‘Contents of clinical dataset: prostate cancer’
Answer: We modified title of tables to suggested template.
Material 2
Along with the URL, there should be some guidance on what one finds on a given URL and how this source could be used.
Answer: We added guidance in Material 2.
Table S11
This table, listed in manuscript on page 16, seems to be missing from the submitted material available for review.
Answer: We clarified Table S11 in Material 2.